# REST Targets JAK–STAT and HIF-1 Signaling Pathways in Human Down Syndrome Brain and Neural Cells

**DOI:** 10.3390/ijms24129980

**Published:** 2023-06-10

**Authors:** Tan Huang, Sharida Fakurazi, Pike-See Cheah, King-Hwa Ling

**Affiliations:** 1Department of Biomedical Sciences, Faculty of Medicine and Health Sciences, Universiti Putra Malaysia, Serdang 43400, Selangor, Malaysia; 2Department of Human Anatomy, Faculty of Medicine and Health Sciences, Universiti Putra Malaysia, Serdang 43400, Selangor, Malaysia; 3Malaysian Research Institute on Ageing (MyAgeing^TM^), Universiti Putra Malaysia, Serdang 43400, Selangor, Malaysia

**Keywords:** down syndrome, rest, organoid, NPC, neuron, astrocyte

## Abstract

Down syndrome (DS) is the most frequently diagnosed chromosomal disorder of chromosome 21 (HSA21) aneuploidy, characterized by intellectual disability and reduced lifespan. The transcription repressor, Repressor Element-1 Silencing Transcription factor (REST), which acts as an epigenetic regulator, is a crucial regulator of neuronal and glial gene expression. In this study, we identified and investigated the role of REST-target genes in human brain tissues, cerebral organoids, and neural cells in Down syndrome. Gene expression datasets generated from healthy controls and DS samples of human brain tissues, cerebral organoids, NPC, neurons, and astrocytes were retrieved from the Gene Ontology (GEO) and Sequence Read Archive (SRA) databases. Differential expression analysis was performed on all datasets to produce differential expression genes (DEGs) between DS and control groups. REST-targeted DEGs were subjected to functional ontologies, pathways, and network analyses. We found that REST-targeted DEGs in DS were enriched for the JAK–STAT and HIF-1 signaling pathways across multiple distinct brain regions, ages, and neural cell types. We also identified REST-targeted DEGs involved in nervous system development, cell differentiation, fatty acid metabolism and inflammation in the DS brain. Based on the findings, we propose REST as the critical regulator and a promising therapeutic target to modulate homeostatic gene expression in the DS brain.

## 1. Introduction

Down syndrome (DS) affects 1 in 800 live births worldwide, and it is the most common chromosomal disorder associated with intellectual disability [1]. DS has many phenotypic features, including smaller brain size, decreased number of neurons, increased astrocytes, abnormal dendrites, abnormal synaptic plasticity, and early neurodegeneration similar to Alzheimer disease (AD) [2,3]. The DS phenotypes may result from the interaction between the overexpression of genes mapping to trisomy chromosome 21 and subsequent dysregulation of genes mapping to different chromosomes [4]. One of the hallmark features of DS is a reduction in brain size and volume. This developmental abnormality occurs throughout the life of individuals with DS, from the fetus to elderly [5,6]. A previous study has confirmed that this pathological change in DS brain is closely related to cognitive, language, and memory impairment [7]. The reduction in neurogenesis is thought to have contributed to decreased DS brain volume and intellectual impairment. Neuronal loss was found in other DS brain regions, including the cerebral cortex, hippocampus, and cerebellum [5,8]. Cellular abnormalities in DS patients are also characterized by increased gliogenesis, especially astrocytogenesis [9]. However, these increased astrocytes have impaired maturation and dysfunction, which may negatively affect neuronal function [9,10].

Overexpression of genes on chromosome 21 is responsible for increased oxidative stress, decreased stress resilience, neurodegenerative diseases, and aging in the DS brain [6,11]. Triplicated genes, such as SOD1, DSCR1, and APP, result in an unbalanced relationship between the generation of free radicals and reactive metabolites, which can lead to oxidative stress and consequent damage to brain cells and tissues [12]. Abnormal brain development and impaired intelligence are common among people with DS, who often develop Alzheimer-like symptoms early in life [13]. Amyloid plaques and tau neurofibrillary tangles are two different forms of brain lesions indicative of AD [14]. These two brain pathological changes are almost common in 40-year-old DS patients, making them more than 90% at risk of developing AD [13,15]. Before the age of 40, AD is rare among individuals with DS, but after that point, its prevalence rises dramatically, reaching 88–100% in patients older than the age 65 [16].

The transcriptional repressor “Repressor Element-1 Silencing Transcription factor (REST)”, often referred to as “Neuronal Restriction Silencing Factor (NRSF)”, is widely known for its involvement in the control of gene expression in neuronal cells. REST is an influential transcription factor that inhibits the transcription of downstream genes by binding to the conserved 23 bp DNA pattern known as repressor element 1 (RE1) [17]. REST is an essential neuronal epigenetic modification regulator that targets genes involved in synaptic function, ion channel conductivity, vesicular transport, and neuronal development [18]. Chromatin immunoprecipitation, combined with deep sequencing (ChIP-seq) technology, discovered 2172 REST-target genes in human embryonic stem cells (ESCs) and 308 REST-target genes in ESC-derived neurons [19,20]. According to the Gene Transcription Regulation Database (GTRD) [21], which provides the most comprehensive set of consistently processed ChIP sequencing data for finding human transcription factor binding locations, there are currently 15,450 REST target genes in the human genome. In a recent study, McGann et al. demonstrated 1735 REST-target genes in the human hippocampus [22]. The REST protein has a high level of expression in ESCs and neural progenitor cells (NPCs) to regulate cellular differentiation and pluripotency [23]. Timely downregulation of nuclear REST in NPC is essential for neurodifferentiation and the development of essential neuronal functions, including axonal growth, synaptic signaling, membrane excitability, and the acquisition of neuronal phenotypes [24]. In mature neurons, REST’s low expression level helps to regulate axonal function and synaptic plasticity [24]. REST also regulates the elaboration of glial cell lineages, gliogenesis, and neuronal–glial interactions [25]. During postnatal brain development, REST fine-tunes genes associated with normal aging processes and synaptic plasticity and enhances neuroprotective effects by suppressing genes implicated in oxidative stress and β-amyloid toxicity [26].

REST, which is active throughout fetal brain development, is reactivated in later life to shield aging neurons from a wide range of stresses, as well as from the damaging effects of the aberrant proteins [27]. In aging humans, elevated levels of REST are linked to the maintenance of cognition and longer lifespan, even when there is Alzheimer disease pathology present [28]. The loss of neuronal REST expression is closely related to a shortened lifespan of neurons [28]. Overexpression of REST upregulates the glutamate transporter protein EAAT2 and protects dopaminergic neurons from excitotoxic damage [25]. Inflammatory mediators, including COX-2, iNOS, IL-1, and IL-6, are produced more frequently, and astrocytes perform pro-inflammatory functions more actively when REST is deficient [29]. In comparison to euploid controls, REST mRNA and protein levels were considerably lower in human fetal DS brains [30]. In trisomy 21 brains, REST expression is lowered by 30% to 60%, and this modification lasts from undifferentiated embryonic stem cells to adult brains [31]. Therefore, altered levels of REST in neural cells may be an essential factor in the DS phenotype, abnormal neurogenesis and gliogenesis, and abnormal cellular function.

The last few decades have seen discoveries in gene expression analysis, which have not only elucidated disease mechanisms at different stages or phenotypes, but they have also enabled gene-targeted therapies and the development of targeted drugs [32,33]. The GEO and SRA are useful, multilingual databases that house information from high-throughput sequencing, microarrays, and other types of genomics and proteomics. In this study, we thoroughly investigated the significance of REST throughout the DS human brain development and function, focusing on REST target genes enrichment in various DS brain tissues (with different developmental stages), cerebral organoids, neural progenitors, neurons, and astrocytes. Our discoveries indicate that REST significantly targets the JAK–STAT and HIF-1signaling pathways in all DS brain tissues, developmental time points, and cell types.

## 2. Results

### 2.1. Identification of DEGs in Down Syndrome Brain Regions and Neural Cells

In this study, we analyzed the DEGs in human brain regions (GSE59630). A total of 1497 DEGs were identified in cerebellar cortex (CBC), 2095 DEGs in dorsolateral prefrontal cortex (DFC), 1092 in hippocampus (HIP), 2396 DEGs in inferior temporal cortex (ITC), 1503 DEGs in prefrontal cortex (OFC), 4743 DEGs in primary somatosensory cortex (S1C), 2173 DEGs in primary visual cortex (V1C), and 1699 DEGs in ventrolateral prefrontal cortex (VFC) (see Appendix A for heat map and volcano plot). Then, the DEGs were further identified in different age groups of human brain tissue (fetal (sixteen to twenty-two gestational weeks), infant (zero to twelve months), child (two to sixteen years), and adult (eighteen to twenty-four years)) (GSE59630). We found 2167 DEGs in the fetal brain, 1940 DEGs in infant brain tissue, 948 DEGs in child brain tissue, and 3912 DEGs in adult tissue (see Appendix A for heat map and volcano plot). In addition, we identified 1385 DEGs in the human brain organoids, 707 DEGs in NPCs, 136 DEGs in neuron cells, and 110 in astrocytes (see Appendix A for heat map and volcano plot).

### 2.2. REST-Targeted DEGs in Down Syndrome

We performed overlapping between REST-target genes and DEGs in human brain regions with DS, presented in Venn diagrams (Figure 1a–h). We verified 1262 (84.30%) REST-targeted DEGs in CBC (representation factor, RF = 1.9 and *p* < 6.567 × 10^−237^), 1721 (82.15%) in DFC (RF = 1.9 and *p* < 2.034 × 10^−298^), 879 (80.49%) in HIP (RF = 1.8 and *p* < 2.178 × 10^−138^), 1981 (82.67%) in ITC (RF = 1.9 and *p* < 0.000 × 10^0^), 1208 (80.37%) in OFC (RF = 1.8 and *p* < 1.461 × 10^−190^), 3945 (83.17%) in S1C (RF = 1.9 and *p* < 0.000 × 10^0^), 1775 (81.68%) in V1C (RF = 1.9 and *p* < 4.537 × 10^−302^), and 1325 (77.99%) in VFC (RF = 1.9 and *p* < 2.025 × 10^−187^). Then, we identified the common genes consistently dysregulated in all regions. Fifty-one common REST-targeted DEGs were obtained, including forty-one up-regulated DEGs and ten down-regulated DEGs in DS (Figure 2).

At the same time, we performed the overlapping between REST-target genes and DEGs in different age groups of brain tissue (Figure 3a–d). We identified 1536 (70.88%) REST-targeted DEGs in fetal brain tissue (RF = 1.6 and *p* < 6.306 × 10^−149^), 1664 (85.77%) in infant brain tissue (RF = 1.9 and *p* < 0.000 × 10^0^), 766 (80.80%) in postnatal brain tissue (RF = 1.8 and *p* < 4.495 × 10^−122^), and 3170 (81.03%) in adult brain tissue (RF = 1.8 and *p* < 0.000 × 10^0^). Of these, we identified the common REST-targeted DEGs that were consistently dysregulated in all age groups. A total of 56 REST-targeted DEGs were obtained, including 39 up-regulated DEGs and 17 down-regulated DEGs in DS (Figure 4).

Furthermore, we performed the enrichment analysis of REST-target genes in DEGs from DS-hiPSC-derived brain organoids and identified a total of 913 (65.92%) REST-targeted DEGs (RF = 1.6 and *p* < 3.608 × 10^−8^) (Figure 5a). In the analyses, based on DEGs of neural cells, we found 482 (68.18%) REST-targeted DEGs in DS NPC (RF = 1.5 and *p* < 5.477 × 10^−39^), 81 (59.56%) in DS neurons (RF = 1.3 and *p* < 2.088 × 10^−4^), and 77 (70.00%) in DS astrocytes (RF = 1.6 and *p* < 3.608 × 10^−8^) (Figure 5b–d).

### 2.3. GO and KEGG Enrichment Analyses of REST-Target DEGs

To further explore the biological process function and mechanism involved in the role of REST in Down syndrome, we performed GO and KEGG enrichment analyses based on the REST-targeted DEGs using DAVID web tools.

In eight human brain regions, the REST-targeted DEGs are highly involved in nervous system development, neuron projection development/maintenance, neurogenesis, cell morphogenesis, apoptotic/cell death, and cell signal transduction (Figure 6a). These genes are mainly involved in the HIF-1, axon guidance, MAPK, fatty acid degradation/metabolism, and Rap1 signaling pathways (Figure 6b). Through analysis of the common REST-targeted DEGs in all regions, we found that the JAK–STAT and HIF-1 signaling pathways were prevalent in all brain regions, and all genes involved in these signaling pathways were upregulated (Figure 2), suggesting a potential loss of REST repression. We summarized the most critical KEGG and GO results involving these regions in Figure 7.

In the fetal brain, the REST-targeted DEGs were represented in ontologies related to the nervous system development, neuron differentiation/development, and neurogenesis. KEGG enrichment results showed that these genes are involved in cellular senescence, as well as pathways of neurodegeneration and apoptosis. In postnatal samples, GO enrichment results of the REST-targeted DEGs relate to processes, such as response to hypoxia, cell morphogenesis, neurogenesis, nervous system development, and axon guidance. The KEGG enrichment analysis revealed that these genes involved the HIF-1 and JAK–STAT signaling pathways and fatty acid degradation/metabolism. In adult samples, the REST-targeted DEGs were enriched for processes related to nervous system development, trans-synaptic signaling, modulation of synaptic transmission, and neurogenesis. The KEGG enrichment further revealed that these genes mainly engage in the HIF-1 and axon guidance signaling pathways and fatty acid degradation/metabolism (Figure 8). The analysis of the common REST-targeted DEGs in all age groups revealed that the Hippo and HIF-1 signaling pathways were prevalent in all age groups (Figure 4). We have summarized the common KEGG and GO results involved in each age group in Figure 9.

The GO enrichment analyses of REST-targeted DEGs in brain organoids revealed processes that were involved in nervous system development, neurogenesis, and neuron differentiation/development (Figure 10a). In contrast, KEGG enrichment results indicated enrichments in the pathways for the cell cycle, axon guidance, and Wnt signaling (Figure 10b).

The REST-targeted DEGs in NPCs were enriched for biological processes related to the nervous system development, neurogenesis, axon development, and cell development/differentiation, which correspond to the KEGG JAK–STAT, Wnt, and axon guidance signaling pathways. The REST-targeted DEGs in neurons were found to be significantly enriched for cell cycle and immuno-inflammatory response processes, corresponding to the KEGG analysis’s JAK–STAT signaling pathway. DS-derived astrocytes (REST) targeted DEGs, and they were mainly involved in cell development/differentiation/migration and fatty acid degradation/metabolism processes. KEGG enrichment analysis revealed that these genes are involved in the JAK–STAT signaling pathway. The top 10 GO and KEGG enrichment results for NPCs, neurons, and astrocytes are presented in Figure 10a,b. We have summarized the common KEGG and GO results in organoids, NPCs, neurons and astrocytes, as shown in Figure 11.

### 2.4. PPI Network Analysis of DEGs and MCODEs

The PPI networks of DEGs and the top three modules in NPCs (see Appendix A for clear original image), neurons, and astrocytes were constructed (Figure 12). We found most of the genes in critical modules were the REST-targeted DEGs, which appeared as red and green, representing up- and down-regulation, respectively, in the modules. Results showed that these genes in the modules in NPCs are involved in cell fate, axon guidance, and neuron projection (Figure 12(e1)). The genes in the modules in neurons were mainly enriched in astrocyte differentiation, glucose metabolism, and mitochondrial function (Figure 12(e2)). Additionally, the genes in the astrocyte modules were primarily enriched in the lipid metabolic process, neuron projection, and axon guidance (Figure 12(e3)).

## 3. Discussion

It is generally believed that dysregulation of REST in the nucleus can disrupt the expression of its target genes. Overexpression of REST will block the expression of downstream genes. In contrast, reduction or loss in REST will diminish or fail to inhibit the expression of downstream genes, leading to their upregulation. In our study, we identified 51 REST-targeted DEGs that were common to all DS human brain regions, of which 41 (80.39%) genes were upregulated, such as IFNAR1, IFNAR2, SLC2A1, NOTCH2, and DONSON. We have also identified 56 REST-targeted DEGs that were common to all time points during DS brain development, of which 39 (69.64%) genes were upregulated, such as IFNAR1, SLC2A1, PFKL, SOD1, and HMGN1. There are 14 genes common to all brain regions and time points of development, 11 of which are upregulated, such as IFNAR1, SLC2A1, BACH1, YAP1, and KDM3A. These REST-target genes are enriched in the JAK–STAT signaling pathway, HIF-1 signaling pathway, nervous system development, cell differentiation, inflammation, and oxidative stress. We also found that REST-targeted DEGs in organoid and neural cells are also enriched for the same signaling pathways and biological processes.

### 3.1. REST and the JAK–STAT Signaling Pathway

Cell membrane receptors, JAKs and STATs, are the three primary protein groups involved in the JAK–STAT signaling pathway. JAK1, JAK2, JAK3, and TYK2 comprise the JAKs, a non-receptor protein tyrosine kinase grouping. STAT1, STAT2, STAT3, STAT4, STAT5, (STAT5A, STAT5B), and STAT6 are members of the STAT family [34]. The JAK–STAT signaling pathway is responsible for cell proliferation, cell differentiation, cell survival, immunity, inflammatory response, and apoptosis [35,36]. It is essential for nervous system development, stem cell maintenance, and gliogenesis. During brain development, activating the JAK–STAT signaling pathway triggers the cell fate switch of NPCs, facilitating their differentiation into astrocytes [37]. Phosphorylation of JAKs (JAK1 and JAK2) enables the activation of the STAT3 transcription factor, which subsequently initiates the expression of astrocyte-specific genes, such as GFAP and S100, thereby specifying the destiny of neural glial cells [37,38]. One of the prominent pathological features in the DS brain is the neurogenesis-to-gliogenesis shift, which results in a decrease in neurons and an increase in astrocytes in DS brains [39]. Although astrocytes play a critical role in supporting neurons with metabolic and neurotrophic processes and regulating essential mechanisms, such as synaptogenesis/plasticity, exocytosis/homeostasis of neurotransmitters, and the cerebrovascular couple [40,41,42], the propensity for astrogliogenesis is not a positive event in DS. Studies have shown that astrocytes exhibit defects in the interlaminar processes, indicating impaired maturation, which leads to dysfunctional astrocytes in DS. Abnormal astrocyte function in the DS brain contributes to mitochondrial dysfunction, increased ROS, and causes apoptosis of neurons [43,44]. A previous study has demonstrated that the JAK–STAT signaling system is activated, contributing to the neurogenic-to-gliogenic shift in DS brains [45]. Many cytokine receptors, such as interleukins, interferons, and growth factors, can trigger the JAK–STAT signaling pathway. In DS, IFNAR1, and IFNAR2, expressions were found to be elevated in human telomerase (hTERT)-immortalized fibroblasts [46]. At nineteen to twenty-one weeks of gestation, IFNAR2 is roughly two-fold more elevated in human fetal DS brains [47]. When IFNAR1 and IFNAR2 are bound to membrane receptors, the JAK–STAT signaling pathway is activated, which triggers the differentiation of NPCs into astrocytes [48,49]. Our study found that IFNAR1, IFNAR2, and OSM, JAK–STAT signaling pathway activators, were up-regulated at least 1.5-fold in DS brains, even though some of these genes were not on chromosome 21. These findings indicate that REST may influence the gene dosage to alter the fate of gliogenesis in DS brains by regulating the JAK–STAT signaling pathway.

In addition to regulating the neurogenic-to-gliogenic shift in the nervous system, JAK–STAT plays a vital role in the inflammatory response in the DS brain. In post mortem human DS brains (less than 40 years old), in contrast to controls of the same age, pro-inflammatory cytokines, including IL-6, TNF-α, and TNF-β, were at least two-fold times greater than controls of the same age [50]. These interleukins and interferons bind to microglial and astrocyte receptors, activating both cell types and inducing the JAK–STAT signaling pathway into astrocytes and microglia. Induced nitric oxide synthase (iNOS) and reactive oxygen species (ROS) generation can be sparked by the activation of the JAK–STAT signaling pathway in reactive glial cells, which can result in neuroinflammation and excitotoxicity [9,51]. Compared to WT astrocytes, glia cell activation was more apparent in the REST knockout PD mice model. The REST-deficit significantly elevated the production of the inflammatory cytokines IL-1, IL-6, COX-2, and iNOS in astrocytes [29]. It has been demonstrated that raising REST expression reduces the production of the mediators of inflammation, including IL-1, IL-6, and IL-10, hence protecting the aging brain [52]. Overexpression of REST in reactive astrocytes upregulates astrocytic EAAT2 to protect neurons from neuronal excitotoxicity and neuroinflammatory damage [25]. However, the JAK/STAT3 signaling pathway was activated, which depressed the uptake of glutamate by astrocytes, leading to neuronal excitotoxicity and neuroinflammation [53]. In animal models of neurodegenerative diseases, the JAK–STAT signaling pathway was activated in reactive astrocytes, and inhibition of the JAK–STAT signaling blocked astrocyte reactivity and reduced neuronal damage [54,55]. Our analysis suggests the loss of REST function leads to de-repression of JAK–STAT, causing its activation and aberrant astrogliogenesis, thereby leading to a series of neuropathological consequences.

The expression of downstream genes crucial for maintaining neural homeostasis is disrupted in pathological situations by dysregulation of REST transcriptional activity [56]. Previous research found that the expression of REST was reduced in DS fetal brain cells [30]. While no research evidence to demonstrate that REST regulates the JAK–STAT signaling pathway directly, a potential link between REST and JAK–STAT can be seen in several neurological disorders. Recent studies have shown that loss of REST in the neural cell nucleus is a key factor in the neuropathology of Parkinson’s Disease (PD) [57], and the JAK–STAT signaling pathway was over-activated in PD [58]. In addition, decreased levels of REST expression were detected in several neuroblastoma cell lines [59,60], and these lines showed excessive JAK–STAT signaling pathway activation [61,62]. However, gene dosage imbalance of DYRK1A in DS disrupts REST protein levels. Over- and under-expression of DYRK1A both resulted in the downregulation of REST in embryonic stem cells [31]. Notably, overexpression of DYRK1A can also activate STAT3 and promote aberrant astrogliogenesis in DS Ts1Cje mice. Therefore, we postulate that reduced or loss-of-REST function may be critical in triggering JAK–STAT and, subsequently, gliogenesis in DS [63].

### 3.2. REST and HIF-1 Signaling Pathway

A transcription factor, known as HIF-1, which is comprised of HIF-1α and HIF-1β, is essential for cells to adapt to hypoxia or low-oxygen environments. Due to the oxygen-dependent HIF proly hydroxylase (PHD), the β subunit is continuously being generated in cells. In contrast, a ubiquitin-dependent protease swiftly destroys the α subunit in normoxic circumstances. In hypoxic conditions, PHD is made inactive, causing HIF-1 to stabilize, and then it forms a complex with HIF-1 in the nucleus, and it ultimately drives the expression of target genes. HIF-1-mediated pathways are critical in cell growth and development, neural stem cell maintenance and differentiation, cell survival, and apoptosis [64,65]. The level and duration of HIF-1 cellular expression under hypoxia must be meticulously balanced to prevent adverse consequences from excessive activity. Under prolonged hypoxia, the HIF-1 is suppressed, necessitating REST to negatively feedback-regulate the HIF-1 mRNA transcriptional activity. REST binds with the HIF-1 promoter in a hypoxia-dependent manner, inhibiting HIF-1 mRNA transcription and promoting HIF-1 protein catabolism in prolonged hypoxia [66]. The expression and transcriptional activity of HIF-1 mRNA and protein were significantly increased after the knockdown or knockout of REST [67].

Hypoxia is a very prevalent phenomenon in individuals with DS, being that 45–50% of them have heart disease [68], 80% of DS children suffer from obstructive sleep apnea [69], and all individuals have smaller-sized airways [70]. Even if they do not have congenital heart disease, newborns with DS experience severe hypoxemia more frequently than the control group [71]. Hypoxia can over-activate the HIF-1 signaling pathway in the absence of REST modulation. HIF-1 activity increases the settings of hypoxic tension or hypoxia, favoring and impairing deliberate NPC differentiation. Activated HIF-1 can alter the fate of NPC and promote its differentiation towards gliogenesis. At the same time, it can lower LIN28A, LIN28B, and HMGA2 expression to prevent neurogenesis [72]. Defective HIF-1 in embryonic stem cells leads to abnormally increased neurogenesis, while stable HIF-1 controls abnormal neurogenesis [73]. Further studies have shown that the HIF-1 pathway can inhibit premature neuronal differentiation independently of the Notch signaling pathway by activating the neuroinhibitory factor Hes1. Thus, dysregulation of REST may be the key factor in the neurogenic-to-gliogenic shift caused by the impaired or dysregulated HIF-1 signaling pathway in DS brains.

The HIF-1 signaling pathway is activated in brain tissues in response to neuronal hypoxia, and upregulated HIF-1 can promote neuronal cell survival [74]. However, it only persists for a short time. The upregulation of HIF-1 will lead to astrocyte activation, which supports neuroinflammatory responses and eventually leads to neuronal cell death [75,76]. Hypoxic circumstances have been found to cause activated astrocytes and microglia to generate IL-1 in a HIF-1-dependent way [77]. A previous study has shown that the neuroprotective effects of HIF-1 are dependent on the regulation of erythropoietin (EPO) production in neurons and astrocytes against hypoxia and apoptosis [78]. According to a study, IL-1β, IL-6, and TNF-α may decrease the level of EPO in astrocytes [79]. However, inhibition of HIF-1α attenuates microglia and astrocyte activation [80]. In the DS brain, activation of the JAK–STAT signaling pathway also has been shown to promote the activation of microglia and astrocytes. Studies have shown crosstalk between the HIF-1 and JAK–STAT signaling pathways. In response to hypoxia in cancer cells, HIF-1 increases the expression of JAK2 and STAT3, activating the JAK–STAT pathway and its downstream signaling pathways.

In contrast, activation of the JAK–STAT pathway, specifically STAT3, increases the expression and stability of HIF-1 [81]. Crosstalk and synergistic effects of STAT3 and HIF-1 signaling pathways have also been observed in several neurological disorders, such as glioma [67] and pericyte glucose deprivation [82]. There seems to be a positive feedback loop between HIF-1 and JAK/STAT3 signaling pathways. Hence, REST may be an important transcription factor in the crosstalk between the JAK–STAT and the HIF-1 signaling pathways that affects astrocyte reactivity. However, the role of the HIF-1 signaling pathway and HIF-1 and JAK–STAT crosstalk in the DS brain remains unclear.

### 3.3. REST Regulates Nervous System Development and Metabolism

Prior research has established that REST plays a role in synaptic function, ion channel conductivity, vesicular transport, and neuronal development. REST is widely expressed in human neural precursor cells, neurons, and glial cells [83]. The expression level of REST protein determines the fate of NPC differentiation into neurons and astrocytes [24,25]. REST is an essential transcription factor in NPCs for neurogenesis and gliogenesis. It is highly expressed in NPCs, and sustained high expression leads to glial cell development. At the same time, REST is naturally decreased in NPCs during late development to induce neuron-specific gene expression to acquire a neuronal phenotype [84]. By suppressing the expression of target genes, REST regulates neuronal epigenomes, neural cell development and differentiation, axon growth, vesicular transportation, ion channel transduction, and synaptic plasticity [85].

Mitochondria play a crucial role in cellular energy production by regulating cellular life and death fate through oxidative stress and integrating signaling networks into several metabolic pathways that control neurogenesis and neuroplasticity [86]. However, a recent study has shown that the upregulation of REST expression in the nucleus protects neurons from oxidative stress and is associated with longevity [28]. Furthermore, the major extrinsic apoptotic pathways are the tumor necrosis factor (TNF) pathway and apoptosis associated with TNF, including the ligand (TRAIL) pathway. TNF and TRAIL, which are both death ligands, bind to their respective death receptors and aid in the recruitment of adaptor proteins (TNFR1-associated death domain and death domain-associated proteins), which help put together the scaffolding for the apoptosis initiation molecules [10]. REST is required to suppress the expression of adaptor proteins, reducing the production of receptor complexes and disrupting downstream apoptotic signaling [28].

This study also found that REST-targeted DEGs were significantly enriched in fatty acid metabolic pathways in childhood and adulthood brains, as well as astrocytes. Metabolic support of neurons by astrocytes depends on neuronal lipid clearance because fatty acid in neuronal lipid droplet is transported to astrocytes in an ApoE isoform-dependent manner [87]. Astrocytes consume fatty acids from stored neurons in lipid droplets by β-oxidation of mitochondria in response to the activity of neurons. This neuronal-astrocyte coupled lipid metabolism mechanism protected neurons from fatty acid toxicity and oxidative stress damage [88]. The relationships among astrocyte lipid metabolism, oxidative stress, energy metabolism, and anti-inflammatory pathways are shown by astrocyte lipid droplet transport and storage. As a result of these pathways being disrupted, the metabolic balance of the central nervous system is altered, resulting in dysregulated energy generation, inflammation, excitotoxicity, toxicity, and pathogenic mechanisms linked to numerous neurodegenerative disorders [89].

## 4. Materials and Methods

### 4.1. Acquisition of DS Transcriptomic Datasets

We downloaded the data for DS human brain tissues and neural cells (NPCs, neurons, and astrocytes) with trisomy 21 from GEO (http://www.ncbi.nlm.nih.gov/geo, accessed on 9 February 2023) [90]. The gene expression datasets were both obtained from the Affymetrix GPL5175 platform with the accession number GSE59630 (human brain tissues) [91], the Affymetrix GPL570 platform with the accession number GSE84887 (NPCs from embryonic stem cells) [92], GSE48611 (neurons from induced pluripotent stem cells) [93], and the Affymetrix GPL6255 platform with the accession number GSE42772 (astrocytes from the human cortex cells) [94]. Based on the platform’s annotation data, we translated probes into their associated gene symbols. In this study, the trisomy 21 human-induced pluripotent stem cell (hiPSC)-derived human brain organoid sequence dataset was downloaded from Sequence Read Archive database (SRA, https://www.ncbi.nlm.nih.gov/sra).

The GSE59630 dataset includes a total of 116 samples from eight brain regions of 58 DS and 58 controls (aged 16 post-conception weeks to 42 years). These samples were grouped according to different brain regions, such as the cerebellar cortex (CBC; ten DSs and ten controls), dorsolateral prefrontal cortex (DFC; eight DSs and twelve controls), hippocampus (HIP; three DSs and three controls), inferior temporal cortex (ITC; five DSs and five controls), prefrontal cortex (OFC; six DSs and six controls), the primary somatosensory cortex (S1C; two DSs and two controls), primary visual cortex (V1C; eleven DSs and eleven controls), and ventrolateral prefrontal cortex (VFC; seven DSs and seven controls). Three healthy controls and five DS human embryonic stem cell (ESC)-derived NPC samples were obtained from the GSE84887 dataset. The GSE48611 dataset comprises six DS hiPSC-derived neuron samples and three from healthy control subjects. The GSE42772 dataset contains primary human astrocytes derived from brain samples of three healthy controls and five DS subjects at 17–20 weeks of gestational age. The hiPSC-derived organoid (days in vitro, DIV 30) dataset was generated from three pairs of DS samples and corresponding isogenic control lines in PRJNA721827 (SRR14244005-SRR14244010).

### 4.2. Identification of Differentially Expressed Genes (DEGs) in DS Samples

For all microarray datasets, the R project was used to identify genes that were differentially expressed between samples of DS and healthy controls. The limma package was used to perform differential expression analysis. For the Next-Generation Sequencing (NGS) dataset, Galaxy Community hub bioinformatics tools (https://usegalaxy.org) were used to perform the analysis. FASTQ files were read and trimmed using Trimmomatic. Then, the transcriptome sequences were aligned using HISAT2, and the hg38 human reference genome was used as the annotation file. Finally, differential gene expression between DS and control was analyzed by using DESeq2. The genes were regarded as DEGs with *p*-value < 0.05 and |log2 FC| > 0.5 for both microarray datasets and NGS data. Using RStudio software, a heatmap and volcanic map were created to display the analysis results (version: 2022.02.2).

### 4.3. REST-Targeted DEGs in Down Syndrome

The REST target genes of humans were obtained from the Gene Transcription Regulation Database (http://gtrd.biouml.org). Then, the overlapping genes between REST target genes and DEGs were illustrated using Venn diagrams (https://bioinfogp.cnb.csic.es/tools/venny/). The representation factor and statistical significance of the overlap between the two sets of genes (human genome as the background) were calculated based on the hypergeometic probability model (http://nemates.org).

### 4.4. Functional Enrichment Analyses for REST-Targeted DEGs and Critical Modules

Gene Ontology (GO) and the Kyoto Encyclopedia of Genes and Genomes (KEGG) enrichment analyses of REST-targeted DEGs were performed using the Bioconductor package clusterProfiler of R to determine the biological processes of REST-targeted DEGs and related pathways. The top 10 GO and KEGG terms with *p*-values < 0.05 were presented.

### 4.5. Protein–Protein Interaction (PPI) Network Analysis of REST-Targeted DEGs and Critical Module

The Search Tool for the Retrieval of Interacting Genes database (STRING; http://string-db.org) was employed to induce the protein–protein interaction (PPI) network [95]. We used an interaction score ≥ 0.4 and employed Cytoscape software (version 3.9.1) to visualize the network. DEGs and REST-targeted DEGs were visualized in different colors. The significant critical modules in the PPI networks were verified by utilizing the Molecular Complex Detection plug-in (MCODE) [96].

## 5. Conclusions

We utilized multiple bioinformatic tools to conclude that REST is critical in DS brain development and neuropathology. Our analysis of differentially expressed genes (DEGs) in DS brain tissues, organoids, and neural cells revealed REST as a significant regulator of gene expression across various brain regions, ages, and cell types. The fact that REST targets many of these DEGs in DS suggests its critical involvement throughout the development of DS neuropathology. GO and KEGG enrichment analyses indicated that REST could serve as a promising therapeutic target by modulating the JAK–STAT signaling pathway, HIF-1 signaling pathway, and astrocyte heterogeneity.

## Figures and Tables

**Figure 1 ijms-24-09980-f001:**
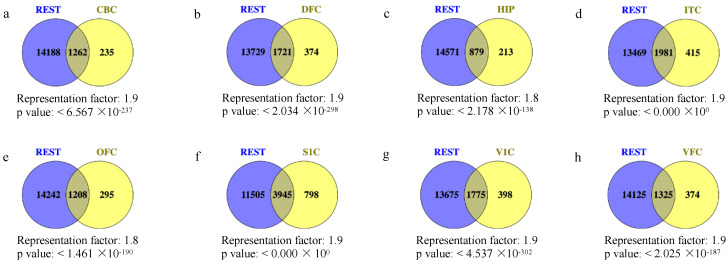
(**a**–**h**) The overlap between REST target genes and DEGs in different brain regions, respectively. The statistical significance of overlapping genes was tested using Fisher’s test, and all *p* values were less than 0.01.

**Figure 2 ijms-24-09980-f002:**
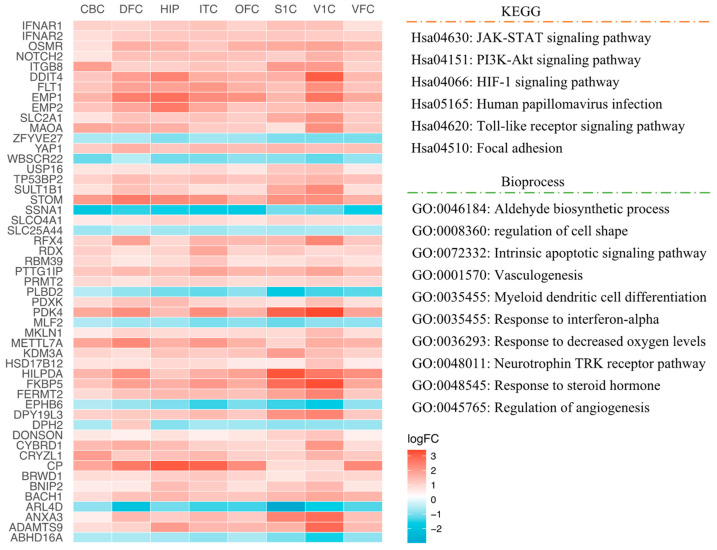
The heatmap shows overlapping genes in different brain regions (CBC, DFC, HIP, ITC, OFC, S1C, V1C, and VFC), as well as the signaling pathways and biological functions involved in these genes.

**Figure 3 ijms-24-09980-f003:**
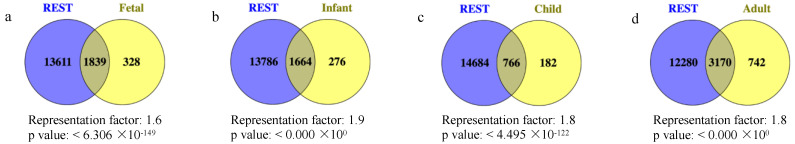
(**a**–**d**) The overlap between REST target genes and DEGs in fetal, infant, child and adult brains, respectively. The statistical significance of overlapping genes was tested using Fisher’s test, and all *p* values were less than 0.01.

**Figure 4 ijms-24-09980-f004:**
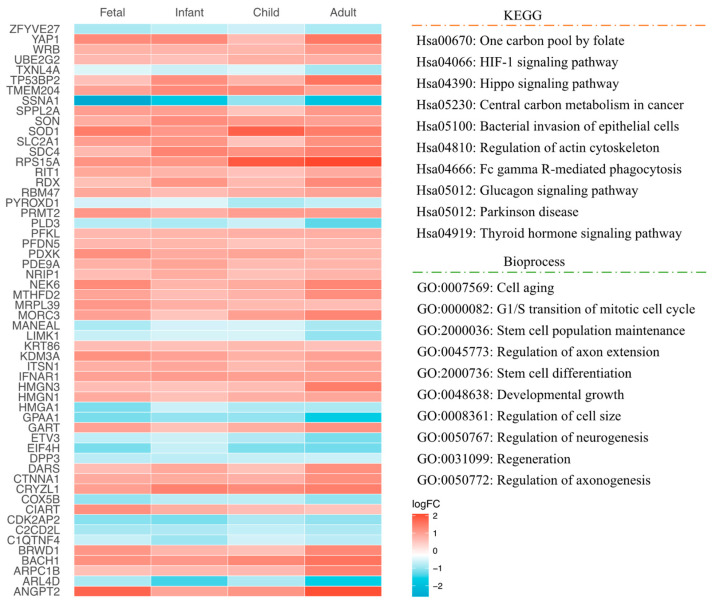
The heatmap shows overlapping genes in different age categories (Fetal, Infant, Child, Adult), as well as the signaling pathways and biological functions involved in these genes.

**Figure 5 ijms-24-09980-f005:**
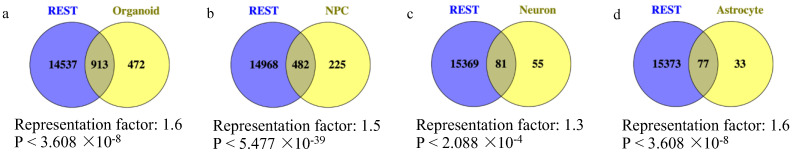
(**a**–**d**) The overlaps between REST target genes and DEGs in organoids, NPCs, neurons, and astrocytes, respectively. The statistical significance of overlapping genes was tested using Fisher’s test, and all *p* values were less than 0.01.

**Figure 6 ijms-24-09980-f006:**
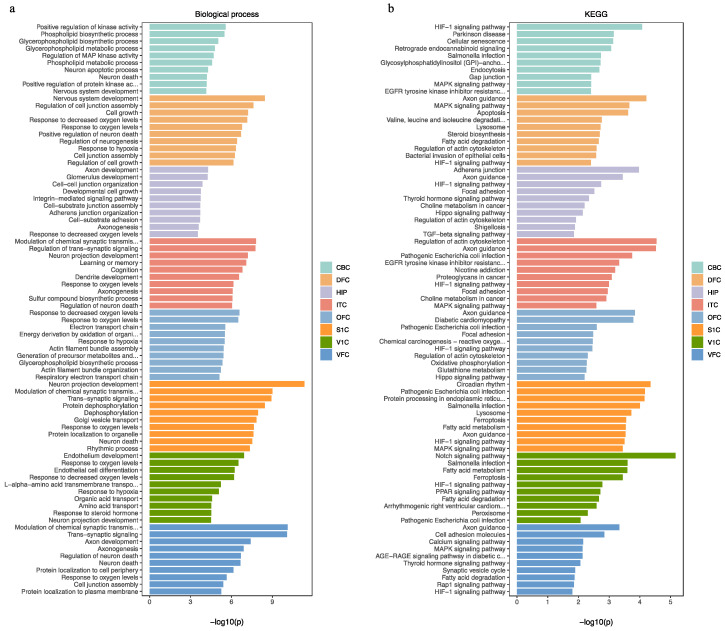
(**a**,**b**) Biological process and KEGG analysis for the REST-targeted DEGs in different brain regions.

**Figure 7 ijms-24-09980-f007:**
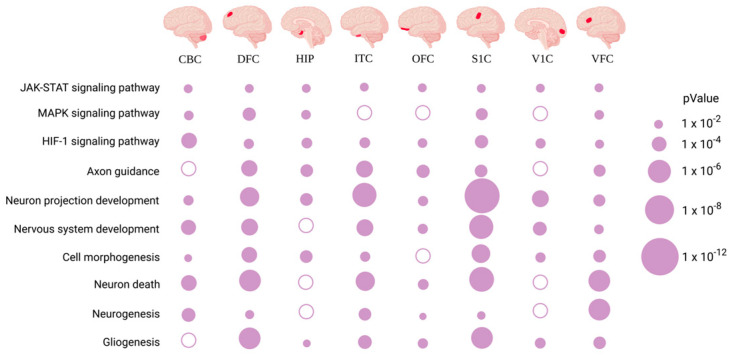
Signaling pathways and biological functions enriched in common REST-targeted DEGs in different brain regions.

**Figure 8 ijms-24-09980-f008:**
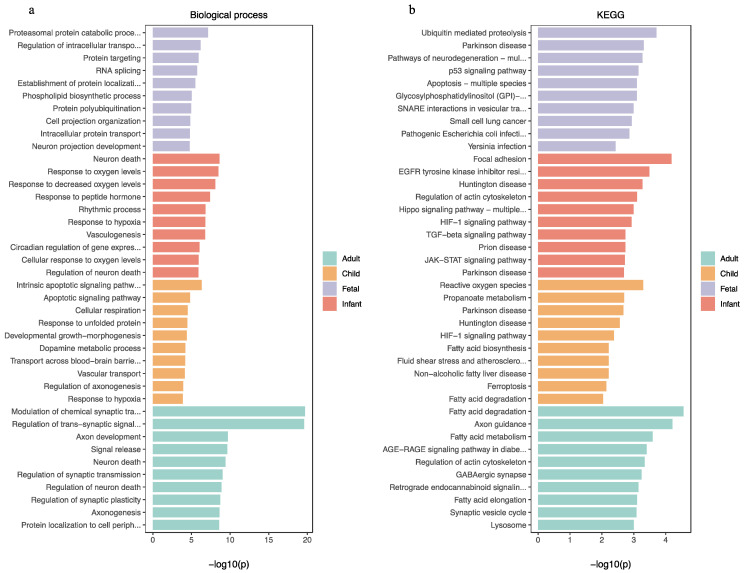
(**a**,**b**) Biological process and KEGG analysis for the REST-targeted DEGs in different ages.

**Figure 9 ijms-24-09980-f009:**
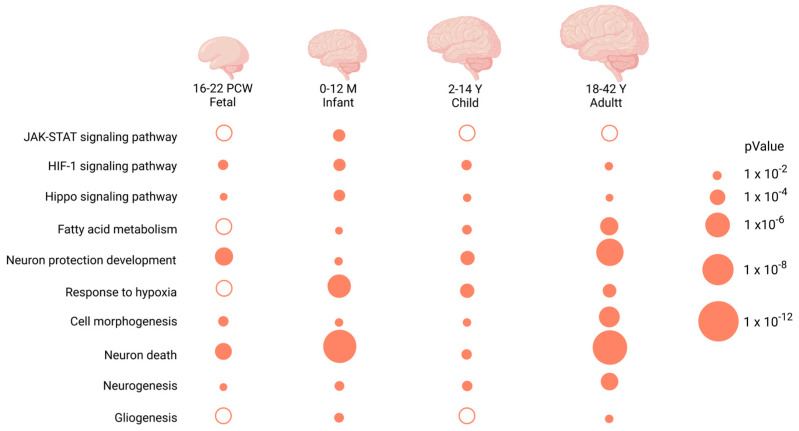
Signaling pathways and biological functions enriched in commonly involved REST-targeted DEGs at different ages (PCW = post-conception weeks, M = month, Y = year).

**Figure 10 ijms-24-09980-f010:**
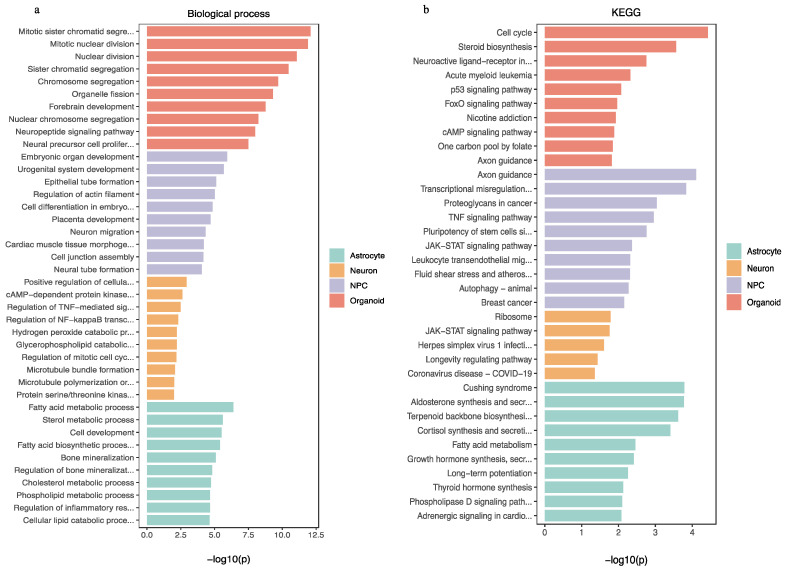
(**a**,**b**) Biological process and KEGG pathway analysis for REST-targeted DEGs in organoids, NPCs, neurons, and astrocytes.

**Figure 11 ijms-24-09980-f011:**
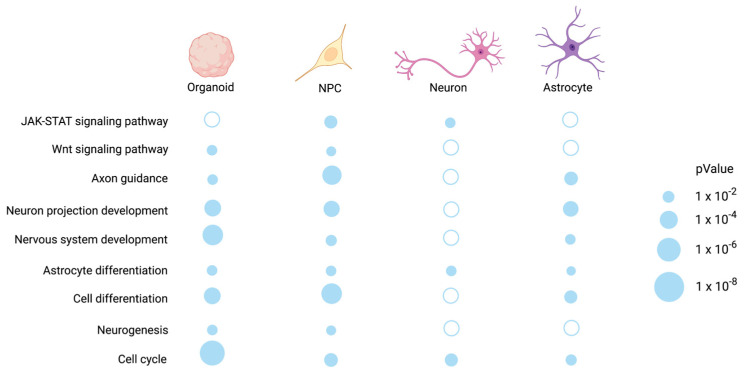
Signaling pathways and biological functions enriched in common REST-targeted DEGs in organoids, NPCs, neurons, and astrocytes.

**Figure 12 ijms-24-09980-f012:**
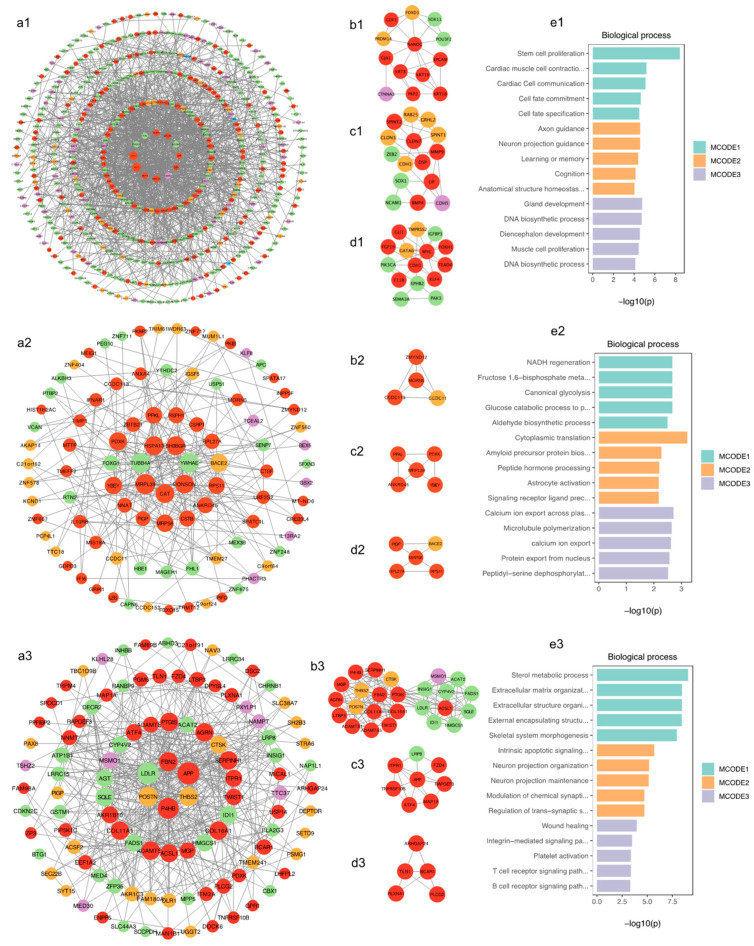
(**a1**–**a3**) The PPI network of DEGs in NPCs, neurons, and astrocytes, respectively, were constructed using Cytoscape. Red represents the REST-targeted upregulated genes, green represents the REST-targeted downregulated genes, yellow represents the non-REST-targeted upregulated genes, and purple represents the non-REST-targeted downregulated genes. The larger the network node degree distribution, the larger the shape. (**b1**–**b3**, **c1**–**c3**, **d1**–**d3**) are the top three models obtained using the MCODE plug-in of Cytoscape in neuron DEGs in NPCs, neurons, and astrocytes, respectively. (**e1**–**e3**) The GO enrichment’s most significant five terms were shown in each top three critical models in neuron DEGs.

## Data Availability

All data sets used in this study are publicly available on the Gene Ex-pression Omnibus (GEO) and the Sequence Read Archive (SRA). The accession numbers are GSE84887, GSE4861, GSE42772, and PRJNA721827 (SRR14244005-SRR14244010).

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
