# Peer review of "REST Targets JAK–STAT and HIF-1 Signaling Pathways in Human Down Syndrome Brain and Neural Cells"

_ijms, 2023, doi:10.3390/ijms24129980_

Round 1

Reviewer 1 Report

Following are my comments for the manuscript:

1) Nicely drafted abstract; there is a typo for spelling of elderly in line 37

2) Very nice an detailed introduction including causes associated with DS, role of REST at different stages of brain development and rationale for current study

3) Well described model selections, Venn diagrams and use of R studio software

4) Nicely described supplemental background data for DEGs described in the beginning of result section 

5) Description of figure 1 i and j is missing in the initial description; though it is described at later stage of manuscript, it becomes little confusing and need to read back and forth; it would be easier if it is elaborated in a chronological manner

6) Overall, nicely described overlapped genes in different brain regions and heatmaps of commonly overlapped genes across regions in figure 1 and 2

7) Description of figure 3e and f is missing in the initial description; though it is described at later stage of manuscript, it becomes little confusing and need to read back and forth; it would be easier if it is elaborated in a chronological manner; overall nice description of overlapped genes with respect to age

8) Well described REST targeted genes in DEGs with respect to DS-hiPSC-derived brain organoids in figure 5

9) Nicely summarized REST targeted genes and signalling pathways in figure 6 ; similar summary with respect to age and function in figure 7 and similar summary of signalling pathways involved in organoids, NPCs, neurons and astrocytes in figure 8

10) Beautiful representation of PPI network analysis in figure 9; there is a typo in figure 9e1 "Stem cell proliferation" 

11) Please elaborate line 366 more clearly; Overall, very nicely written discussion with relevance of REST levels and JAK/STAT as well as HIF-1 alpha pathways and its association with nervous system development and metabolism. Great work!!

Author Response

Dear reviewer,

Thank you for your useful comment. We revised the manuscript according to your suggestions.

Thank you.

Best regards,

Tan Huang.

Reviewer 2 Report

This manuscript describes the role of REST as a signaling molecule in brains and neural cells of Down's Syndrome (DS). The manuscript is well written and the data is in sync with the overall theme of the paper. Below are some of my comments for the paper.

1) Fig 8. Were the comparisons in Fig 8 only of healthy samples or even the DS samples? That part was not very clear to me in the figure. Could you please elaborate on that figure?

Author Response

Dear reviewer,

Thank you for your useful comment. We revised the manuscript and response the comment.

Thank you.

Best regards,

Tan Huang.
